Beyond labs: unveiling dynamics of dental students’ transition from pre-clinical to clinical training in a Saudi dental school

Khattak Osama 1
http://orcid.org/0000-0002-3178-9513 Ganji Kiran Kumar 2 3 4 kiranperio@gmail.com
Iqbal Azhar 1
http://orcid.org/0009-0000-6363-8958 Altassan Mosa 5
Khan Farida Habib 6
Anis Rabia 7
1 Department of Restorative Dental Sciences, Jouf University , Sakaka, Aljouf , Saudi Arabia
2 Department of Preventive Dentistry, Jouf University , Sakaka, Aljouf , Saudi Arabia
3 Department of Dental Research at Center for Global Health Research, SMC, Saveetha Institute of Medical & Technical Sciences , Chennai, Tamilnadu , India
4 Institute of Health Professions Education, Sri Balaji Vidyapeeth , Pondicherry, Tamilnadu , India
5 Department of Oral and Maxillofacial Prosthodontics, King Abdul Aziz University , Jeddah, Jeddah , Saudi Arabia
6 Department of Family and Community Medicine, University of Hail , Hail , Saudi Arabia
7 Department of Oral and Maxillofacial Surgery, Isra University , Hyderabad , Pakistan
Abu Hasna Amjad
Electronic publication date: 2024 Sep 10
Publication date: 2024
Volume: 12
Electronic Location ID: e18019
Received 2023 Dec 29; Accepted 2024 Aug 9
Copyright: © 2024 Khattak et al.
Copyright year: 2024
Copyright holder: Khattak et al.
License: This is an open access article distributed under the terms of the Creative Commons Attribution License, which permits unrestricted use, distribution, reproduction and adaptation in any medium and for any purpose provided that it is properly attributed. For attribution, the original author(s), title, publication source (PeerJ) and either DOI or URL of the article must be cited.
License URL: https://creativecommons.org/licenses/by/4.0/

Keywords: Dental students, Preclinical, Dental skills, Transition, Clinical

Funding: Deanship of Scientific Research at Jouf University This work was funded by the Deanship of Scientific Research at Jouf University through the Fast-track Research Funding Program. The funders had no role in study design, data collection and analysis, decision to publish, or preparation of the manuscript.

==============================
Objective

To assess the factors affecting the transition of dental students from pre-clinical to clinical courses in an outcome-based curriculum.

Methods

This cross-sectional study surveyed dental students in the third and fourth academic years of the Bachelor of Dental and Oral Surgery (BDS) program at the College of Dentistry, Jouf University. Ethically approved and powered by the G Power software, the study employed a modified questionnaire validated through a pilot test to assess five domains. Likert scale responses were analyzed using SPSS v.25, revealing insights into clinical workload, patient interaction, and learning experiences. Multiple regression analysis was used to assess the impact of clinical skill application, workload, transition to clinics, and patient interaction on learning experience as well as CGPA. The Mann-Whitney U test compared the ranks of two independent samples, making it less sensitive to outliers and more suitable for data with non-normal distributions.

Results

In this study, the response rate of the participants was 70%. A total of 44 dental students in their third and fourth years of the program completed the survey. The multiple regression analysis showed that the predictors collectively explained 36.1% of the variance in the learning experience (Adjusted R2 = 0.361). “Transition to Clinics” had a significant positive effect on learning experience (β = 0.292, p = 0.012), “Workload” (β = −0.203, p = 0.393) and “Patient Interaction” (β = 0.443, p = 0.168) were not significant predictors. The Mann-Whitney U test revealed no significant gender differences in transition to clinics, workload, patient interaction, application of clinical skills, and learning experience (U = 33.09 to −40.33, p > 0.05), but a significant difference in transition to clinics between third- and fourth-year students (U = 31.56 to −43.24, p < 0.05).

Conclusion

The results of this study demonstrate that the transition to clinical training can be intricate, and that multiple elements have an impact on this process. It is crucial to have support systems that facilitate the transition into the clinical learning environment.

Introduction

The training of dental students involves the integration of knowledge and skills in a decision-making context, enabling them to perform independently as future clinicians to manage diverse clinical cases and deliver premium care to their patients (Serrano et al., 2018). The dental education process consists of several stages, the first of which is theoretical, followed by simulation, and finally the clinical phase in which students are expected to work on patients. Globally, the transition period between pre-clinical to clinical education occurs after the completion of preclinical dental skill courses (Frese et al., 2018). The pre-clinical-to-clinical transition refers to the stage in which students shift from being solely taught, to acquiring the responsibility of patient care while fulfilling their academic obligations. This transition is influenced by three domains: (1) teaching factor, which encompasses the educational content, integration of skills, and further development of clinical competencies, (2) student factor, which involves the cognitive and affective experiences of students during this transition, and (3) learning environment (LE), which connects the first two domains (Serrano et al., 2021).

Evidence regarding dental education indicates that several obstacles arise from the transition from pre-clinical to clinical training, such as difficulty in adapting to professional socialization and interprofessional collaboration, an increase in workload, lack of confidence in applying the theoretical knowledge and skills to real-world patients, and ongoing learning and assessments (Malau-Aduli et al., 2022). Undergraduate dental students must quickly acquire both cognitive and psychosocial skills in a short period of time, not only for patient care, but also for working with clinical teachers. Moreover, they must promptly convert the practical skills learned during preclinical education into effective clinical performance. This challenging task becomes more difficult owing to additional requirements such as managing patients, hygiene guidelines, and documentation of patient records. These factors have a psychosocial impact and cause added stress, which can affect learning, particularly during the first weeks of clinical treatment (Frese et al., 2018; Malau-Aduli et al., 2022). The effects of this gap on students include an abrupt decline in the expected performance in clinics, errors when administering treatment to patients, and an overall lack of attributes required to effectively collaborate with other members of a healthcare team. Transitioning from pre-clinical to clinical training can be a significant challenge for dental students, as it involves not only a shift in the learning environment, but also in responsibilities and expectations. Understanding this transition from an educational and psychological perspective can provide valuable insights into the factors that influence student experiences. Malcolm Knowles’ theory of andragogy (Knowles, 1984) emphasizes that adult learners are self-directed and learn best when the learning is relevant to their personal and professional goals. During the transition to clinical training, students may benefit from opportunities for self-directed learning, such as case-based learning or problem-based learning, in which they can apply theoretical knowledge to real-life clinical scenarios.

Clinical transition in dentistry has been a topic of debate for several years. The correlation between the performance in simulation laboratories and real-life environments was found to be weak (Arigbede, Denloye & Dosumu, 2014). No correlation was found between the results of an examination for preparing a typodont intended to assess the clinical skills of students, and those of a clinical competency exam that involved the preparation of a full crown (Curtis et al., 2007). This suggests that the performance in simulation laboratories may not be an accurate predictor of clinical performance, indicating that students who are unable to transfer the acquired skills in the clinical context feel stressed and pressured to perform. The perceived stress of students at the start of their clinical dentistry training evidently affected their clinical performance. Clinical years are the most stressful years in undergraduate dental curricula. The pre-clinical-to-clinical transition in dentistry has been identified as a source of stress for students (Alzahem et al., 2011).

Numerous studies have been conducted to closely examine this transition gap; however, the results remain contradictory and largely inconsistent owing to the diverse geographical settings, divergent variables, non-uniform data samples, and the rapidly changing definition of the ideal LE in terms of the student perceptions (Halboub et al., 2018). Additionally, there are insufficient follow-up studies regarding the suggested changes from previous literature to provide students with the credence needed for their widespread application. Despite the initial positive preclinical evaluation, the performance of most dental students has been demonstrated to significantly decline in clinical practice. The dental program training at Jouf University, Kingdom of Saudi Arabia spans over 5 years plus a year internship. The pre-clinical course is during the second year of the program and is a prerequisite for clinical courses. The students transition from the preclinical stage to the clinical stage at the end of the second year. This study aimed to assess the factors affecting the transition of dental students from preclinical to clinical courses in an outcome-based curriculum. This will provide useful insights into how students perceive this transition phase and identify areas that can be improved for better adaptation to the new learning environment to facilitate a smooth transition. It will also help teaching faculty establish evidence-based strategies to make essential changes to the curriculum, clinical setup, and learning environment of the university setting. Five important domains were selected to identify gaps affecting the process: transition to clinics, workload, patient interaction, application of clinical skills, and learning.

Methods

Study design

This cross-sectional descriptive study was conducted at the College of Dentistry, Jouf University, which included the collection and analysis of quantitative data through a survey conducted at the end of the academic year (2022–23) of the Bachelor of Dental & Oral Surgery (BDS) program, allowing students to reflect on their experiences of previously completed courses. This also enabled them to provide information regarding their transitions and experiences during their academic years. The ethical approval of this study was obtained from the Local Committee for Bioethics of the Jouf University (2-03-44).

Participants

The sample size was determined using G * Power software. The power of this study was calculated by using the estimated effect size, which was based on findings from a similar study performed previously and was considered as 0.5 (medium) (Drovandi et al., 2020). The sample size was calculated with p < 0.05 and a 95% confidence interval using the G * power software (Faul et al., 2007). The students who had completed the pre-clinical courses and registered for the clinical courses were included in the study. Students who did not meet these criteria were excluded from this study. The study population comprised 65 students in their third and fourth year of the BDS program who fulfilled the inclusion criteria. Two students dropped out of the program; hence, 63 students completed the clinical course and were invited to participate in the study. The study population included a mix of students, including those who recently completed the preclinical course (third-year students) and those who completed the courses nearly 1 year ago (fourth year). This helped obtain widespread data for the study, which included a heterogeneous population in which students from various years of the program experienced the same transition process in similar settings at different times.

Data collection

Students were invited to participate through e-mail and announcements on Blackboard (a virtual learning environment). Students who agreed to participate in the study were provided informed consent, stating the details of the study. The students were enrolled in this study only after they provided their informed consent. Participants were assured that their participation would have no negative effects. This questionnaire was adapted from the study by Prince et al. (2005). The questionnaire comprised of questions related to the transition related problems. This was modified for dental students by two faculty members from the Department of Restorative Dentistry and validated through a pilot study. The five areas identified were transition to clinics, clinical workload, patient interaction, application of clinical skills, and learning experience. These five areas were identified as the ones which covered the problem areas of the transition process. The tool consisted of 51 questions which fell under the five mentioned themes. The responses were based on a Likert scale (1 = strongly agree to 5 = strongly disagree). The modified questionnaire was administered to 10 students as a pilot and re-administered to the same students after 2 weeks to check its reliability using Cronbach’s alpha test, which was found as 0.85 The validated questionnaire was then distributed to all participating students using Google Forms. The questionnaire assessed five major domains: Transition to clinics, clinical workload, patient interaction, application of clinical skills, and learning experience.

Data analysis

The data were imported into Microsoft Excel, and statistical analyses were performed using the Statistical Package for the Social Sciences (SPSS) (IBM Corp., Version 25.0. Armonk, NY: USA). Descriptive statistical analysis was conducted to calculate the mean and frequency of ratings. Multiple regression analysis was used to assess the impact of clinical skill application, workload, transition to clinics, and patient interaction on learning experience as well as CGPA. The Kolmogorov-Smirnova test revealed that data of responses data do not follow a normal distribution (p < 0.05) hence Mann-Whitney U test was used to assess whether the two-sample means were equal, that is, mean responses obtained from males and females as well as third- and fourth-year groups. The Mann-Whitney U test compared the ranks of two independent samples, making it less sensitive to outliers and more suitable for data with non-normal distributions.

Results

This section presents the findings of our study on the transition from pre-clinical to clinical training in dental education. The analysis covers the relationships between the learning experience and various predictors, including clinical skills application, workload, transition smoothness, and patient interaction. A total of 44 students participated in the survey, with a response rate of 70%. The sample comprised 58% females and 42% males. Of the student participants, 60.6% (26 students) strongly felt that orientation should be provided at the start of clinical practice, 49.3% (22 students) strongly felt they were working as dentists for the first time, 36.6% (16 students) strongly agreed that working with peers was easy, the first week was easier than anticipated, and that they were prepared for clinical training, and 38.6% (17 students) strongly agreed that they needed time to adjust to the new environment (Table 1). Of the student participants, 70.5% (31 students) felt that there was a heavy workload in the clinics, which significantly increased after transitioning to the clinics (Table 2). Of the student participants, 80% (35 students) felt that the interaction with real patients forced them to study the relevant literature. A total of 35.2% of the students strongly felt that they would have preferred patient interaction earlier in the program, whereas 26.8% were uncertain about the same issue. A total of 23.9% of the students felt uncomfortable while examining patients, whereas a similar percentage of students were uncertain when asked the same question (Table 3). Of the students, 50% agreed that clinical faculty supervised them in clinics; students had sufficient relevant basic and clinical knowledge and skills for application in clinical settings. Furthermore, 35.2% of the students felt that the knowledge acquired at this stage was different from what they needed to perform in the clinic. A total of 36.6 % of the students were unsure whether their knowledge was sufficient to perform in the clinic (Table 4). Furthermore, 80% of the students felt that there was sufficient chairside learning in clinics and that retaining clinically acquired knowledge was easy for them; 70% of the students indicated that they had studied and passed the exams. More than half (52%) of the students indicated that they studied the problems encountered in clinics and that their learning was influenced by the questions presented by the clinical faculty. A total of 46.5% of the students indicated that they began to study more after starting clinical practice; 70% of the students indicated that pre-clinical courses helped them prepare sufficiently for clinics (Table 5).

Table 1 Frequency distribution of responses with regards to transition to clinics (n = 44).

Components	Strongly disagree	Disagree	Neutral	Agree	Strongly agree	
%	%	%	%	%	
Orientation should be given to all dentistry students starting the clinics	–	–	1.4	38	60.6	
The first few weeks of clinical work were better than expected	–	7	11.3	45.1	36.6	
Working with my fellow student was easy	–	7	11.3	45.1	36.6	
I needed time to adjust to the new environment		4.2	14.1	43.7	38.6	
This was the first time I experienced what it is like to work as a dentist	2.8	4.2	15.5	28.2	49.3	
I experienced a great deal of stress	–	2.8	19.7	39.4	38	
The first few weeks working in the clinics were difficult for me	2.8	5.6	22.5	40.8	28.2	
l experienced an abrupt transition from preclinical to clinical training	1.4	4.2	21.1	45.1	28.2	
The clinical staff provided sufficient support	1.4	11.3	19.7	36.6	31	
The transition from preclinical to clinical training was smooth	1.4	5.6	33.8	28.2	31	
I felt prepared for the clinical training	4.2	11.3	21.1	33.8	29.6	
I was very uncertain at the beginning of the clinical placements	4.2	7	28.2	29.6	31	
I felt prepared for clinical training	4.2	7	19.7	32.4	36.6	

Table 2 Frequency distribution of responses with regards to workload (n = 44).

Components	Strongly disagree	Disagree	Neutral	Agree	Strongly agree	
%	%	%	%	%	
The workload during the clinics is heavy	1.4	7	19.7	40.8	31	
There is a significant difference between my workload before and after the transition into the clinical training	1.4	1.4	21.1	45.1	31	
Clinical work hours are very long	5.6	14.1	28.2	26.8	25.4	
It was difficult getting used to the work routine	4.2	11.3	23.9	33.8	26.8	
I get sufficient time to study during my clinical rotations	4.2	14.1	32.4	32.4	16.9	
Clinical workload is tiring	2.8	2.8	32.4	38	23.9	

Table 3 Frequency distribution of responses with regards to patient interaction (n = 44).

Components	Strongly disagree	Disagree	Neutral	Agree	Strongly agree	
%	%	%	%	%	
Contact with real patients stimulates me to study	–	4.2	16.9	43.7	35.2	
My first contact with real patients was during the first semester of third year	–	7	15.5	42.3	35.2	
I would have liked real patient contact earlier in the dental program	1.4	8.5	26.8	28.2	35.2	
I feel patients feel uncomfortable when they are examined by a student	7	12.7	18.3	36.6	25.4	
I feel uncomfortable when I examine a patient.	15.5	19.7	23.9	23.9	16.9	

Table 4 Frequency distribution of responses with regards to clinical skill application (n = 44).

Components	Strongly disagree	Disagree	Neutral	Agree	Strongly agree	
%	%	%	%	%	
Clinical faculty supervises while I examine the patient	1.4	7	23.9	46.5	21.1	
I can take history of the patient	–	–	22.5	39.4	38	
I can do an oral health assessment	2.8	5.6	16.9	39.4	35.2	
I can communicate comfortably with the patient	–	1.4	16.9	35.2	46.5	
I can apply knowledge in practice	–	2.8	12.7	42.3	42.3	
The knowledge acquired during preclinical training is applied in the clinical phase	–	2.8	18.3	46.5	32.4	
I feel there are gaps in my knowledge	1.4	8.5	29.6	33.8	26.8	
I have sufficient basic science knowledge	1.4	1.4	22.5	46.5	26.8	
I am prepared to perform clinical skills	–	1.4	19.7	45.1	33.8	
The knowledge required in clinical practice is different from my theoretical knowledge	1.4	11.3	22.5	35.2	29.6	
I have sufficient clinical science knowledge.	–	1.4	25.4	50.7	22.5	
I have sufficient behavioral science knowledge.	–	2.8	31	43.7	22.5	
The level of my knowledge is sufficient	–	5.6	36.6	38	19.7	
I felt prepared for clinical skill performance	1.4	4.2	23.9	39.4	31	

Table 5 Frequency distribution of responses with regards to learning (n = 44).

Components	Strongly disagree	Disagree	Neutral	Agree	Strongly agree	
%	%	%	%	%	
There is a lot of learning from chairside teaching	1.4	2.8	12.7	39.4	43.7	
The knowledge acquired in clinical is easier to remember	–	2.8	15.5	39.4	42.3	
Clinical faculty are good teachers	–	2.8	18.3	40.8	38	
I am able to judge my own progress	–	5.6	16.9	45.1	32.4	
I study primarily to pass tests and examinations	2.8	9.9	16.9	33.8	36.6	
What I study depends on the problems I face during my clinical exposure	1.4	2.8	25.4	52.1	18.3	
Problem based learning (PBL) provided good preparation for clinical practice	7	7	29.6	38	18.3	
I study more after starting my clinical training	2.8	7	16.9	46.5	26.8	
My learning is influenced by the questions from clinical faculty	2.8	4.2	23.9	50.7	18.3	
I feel the need to study because I have forgotten my theoretical knowledge	4.2	4.2	26.8	32.4	32.4	
Pre-clinical courses were good preparation for learning skills which are applied in the clinics	–	4.2	23.9	40.8	29.6	

Regression analysis: Table 6 presents the detailed regression coefficients for each predictor variable. As shown, the ‘Transition to Clinics’ had the highest standardised coefficient, emphasizing its importance in the student’s learning experience. The multiple regression analysis showed that the predictors collectively explained 36.1% of the variance in the learning experience (adjusted R2 = 0.361). The overall model was significant (F = 1.34, p < 0.05). The R value of 0.630 indicates a moderately positive relationship. The Durbin-Watson statistic of 1.850 suggests no significant autocorrelation, ensuring the reliability of our regression model. Among the predictors, ‘Transition to Clinics’ (Table 7) had a significant positive effect on learning experience (β = 0.292, p = 0.012), indicating that smoother transitions were associated with better learning outcomes. However, ‘Workload’ (β = −0.203, p = 0.393) and ‘Patient Interaction’ (β = 0.443, p = 0.168) were not significant predictors, suggesting that other factors may be more critical in influencing the overall learning experience. Table 8 summarizes the multiple regression analysis for cumulative GPA, indicating that the predictors are not strong determinants of GPA, with low R2 value. The “Transition to Clinics” variable had an unstandardized coefficient (B) of 0.073 (SE = 0.437), indicating a positive association with CGPA; however, this was not significant, t = 0.167, p = 0.868. The model had an R2 of 0.051 and an F value of 0.70. “Workload” had a negative coefficient (B = −0.203, SE = 0.236), but was not significant, t = −0.861, p = 0.393. “Patient Interaction” showed a positive association (B = 0.443, SE = 0.318), yet it was not significant, t = 1.393, p = 0.168. “Clinical Skill Application” had a negligible effect (B = −0.002, SE = 0.476) and was not significant, t = −0.004, p = 0.997. “Learning Experience” had a slight negative association (B = −0.062, SE = 0.375), but was not significant, t = −0.165, p = 0.869. Overall, none of the predictors showed significant relationships with CGPA, and the model explained a small portion of the variance in CGPA.

Table 6 Multiple regression analysis for learning experience.

Model	R	R2	Adjusted R2	Std. Error of the estimate	Durbin-Watson	
Clinical skill application, workload, transition to clinics–patient interaction learning experience	0.630	0.397	0.361	0.35986	1.850	

Table 7 Summary of multiple regression analysis with odd ratio for learning experience.

Parameters	Unstandardized coefficients	Standardized coefficients	t	Sig.	Collinearity statistics	
B	Std. Error	Beta	Tolerance	VIF	
Learning experience-transition to clinics	0.354	0.137	0.292	2.588	0.012	0.720	1.390	
Learning experience-workload	0.038	0.077	0.053	0.489	0.626	0.784	1.275	
Learning experience-patient interaction	0.018	0.104	0.022	0.172	0.864	0.534	1.872	
Learning experience-clinical skill application	0.510	0.143	0.445	3.568	0.001	0.588	1.700	

Table 8 Summary of multiple regression analysis with odd ratio for cumulative gross point average.

Parameters	Unstandardized coefficients	R square	F value	t	Sig.	
B	Std. Error	
CGPA-transition to clinics	0.073	0.437	0.051	0.70	0.167	0.868	
CGPA-workload	−0.203	0.236	−0.861	0.393	
CGPA-patient interaction	0.443	0.318	1.393	0.168	
CGPA-clinical skill application	−0.002	0.476	−0.004	0.997	
CGPA-learning experience	−0.062	0.375	−0.165	0.869	

The U-value in the Mann-Whitney U test was 33.09 to 40.33 (p > 0.05) indicating that there were no significant differences in the variables of transition to clinics, workload, patient interaction, application of clinical skills, and learning experience between male and female students (Fig. 1). The U-value in the Mann-Whitney U test was 31.56 to 43.24 (p < 0.05) indicating a significant difference in the variables: transition to clinics between third- and fourth-year students (Fig. 2).

Figure 1 Comparison of mean rank responses for variables among male and female students.

Figure 2 Comparison of mean rank responses for variables among third and fourth year students.

Discussion

This study highlights the academic journey of dental students during their transition to a clinical setting, thereby pinpointing areas for improvement in the educational process. Dental students’ perceptions were measured considering the transition from preclinical dental courses to clinical settings at the College of Dentistry, Jouf University. The findings of this study provide important insights into the challenges and experiences of dental students during their transition from preclinical to clinical training. The literature demonstrates that these insights can have significant implications for dental education and can inform of interventions and strategies to enhance the clinical training experience of dental students (Godefrooij, Diemers & Scherpbier, 2010).

A key finding of this study is the importance of proper orientation programs at the start of clinical training. More than 60% of the students expressed the need for better preparation and guidance during this transition phase (Table 1), which aligns with previous studies emphasizing the importance of orientation programs to help students adjust to clinical environments (Javed et al., 2023; Montero et al., 2018). This highlights the need for dental institutions to invest in robust orientation programs that provide students with a clear understanding of clinical expectations, workflows, and resources, and address any anxieties or uncertainties that students may have during this transition period. This study also found that nearly half of the participants (50%) indicated it was their first time they felt that they were working as “dentists.” This finding highlights the importance of early clinical exposure in dental programs, which was also highlighted in the literature (Ali et al., 2018). Providing students with exposure in the early stages of the program will help them settle in the new learning environment, which is the “clinic” in this scenario. Early orientation can offer benefits such as applying the knowledge gained, linking fundamental theoretical concepts to practical clinical contexts, development of teamwork, and interpersonal and communication skills before the start of clinical training.

Another finding of this study was that students felt prepared while transitioning from skilled laboratories to clinics. This reinforces the effectiveness of an integrated curriculum in which pre-clinical courses prepare students for the skills required at the clinical stage. This is similar to previous studies in which the integration of basic and clinical sciences helped improve learning (van der Hoeven et al., 2020). The findings of this study emphasize the importance of the guidance required for students in dental clinics, such as managing the clinical workload and assigning tasks to students, which can be attributed to the lack of a clinical mentorship program in the study setting. A previous study by Moore et al. (2021) highlighted the importance of mentoring programs in the undergraduate dental curriculum. The clinical mentor can provide guidance in clinics and allocate patients to new dentists while considering their level of experience and complexity of the patient case. This will help in the gradual increase of the cognitive load, with easier tasks performed earlier and complex tasks later on. More than 70% of dental students perceived the clinical workload as excessive (Table 2), with a significant increase in workload after transitioning to clinical training. This finding is consistent with those of previous studies that reported high levels of stress and workload among dental students during clinical training (Divaris et al., 2008). Heavy workloads in clinics may negatively impact student time management skills, stress levels, and ability to balance academic and clinical responsibilities. The adverse effects of an enhanced workload on students during their transition to clinical studies were clearly demonstrated by the quantitative results. This finding is consistent with those of previous studies (Prince et al., 2005). High levels of perceived workload may result in cognitive overload, in which students are overwhelmed by an excessive amount of information to process (Van Merriënboer & Sweller, 2005).

Dental educators must acknowledge and address these challenges by providing appropriate support systems, resources, and time management training to help students effectively manage their workloads during clinical training. Thus, students can be guided through clinical mentoring programs, which has been previously highlighted in literature as well (Moore et al., 2021). Patient interaction was also found to be a significant factor in the perception of dental students. While many students reported that interacting with actual patients helped them apply their knowledge and study relevant literature (nearly 80%, Table 3), a considerable proportion felt that they were uncomfortable being examined by other students, and others felt uncomfortable while examining patients. These findings highlight the need for a comprehensive training regarding patient communication and interpersonal skills for dental students to ensure effective and professional interaction with patients (Jones et al., 2017). Dental educators should prioritize the development of communication skills and professionalism in their curriculum to ensure that students are sufficiently prepared to interact with patients in a compassionate, ethical, and competent manner.

The study also revealed that the application of clinical skills was positively perceived by students, with many reporting that they were supervised by the clinical faculty and had sufficient relevant basic and clinical knowledge. Dental skills required to manage patients have evolved over time (van der Hoeven et al., 2020). Certain contemporary skills required in clinical practice are not incorporated in the dental courses. This was also reported by some students in this study (Table 4). This finding highlights the importance of aligning pre-clinical and clinical curricula to ensure a smooth transition and adequate preparation of students for clinical practice (Montero et al., 2018). Dental educators should carefully review the content and sequence of their curricula to ensure that students are provided with the necessary knowledge and skills to effectively apply their learning in clinical settings. The clinical faculty involved in the teaching of skills at the pre-clinical stage can help students adapt to a new learning environment. The results of this study demonstrate that the perception of students of the clinical staff, chairside teaching, and teaching of the clinical skills are positive, which is similar to findings previously reported in literature (Gerzina, McLean & Fairley, 2005). The findings also revealed that pre-clinical courses sufficiently prepared students to perform procedural skills in dental clinics. This differs from a recent study regarding the preparedness of dental students, which demonstrated that students were not confident regarding their preparedness during the clinical stage (McNutt et al., 2023). A study conducted at the Cardiff Dental School reported a lack of confidence among students in performing complex dental skills (Gilmour et al., 2016). This difference can be attributed to the fact that most of the participants in our study were in the first clinical year of the dental program and had minimal exposure to clinics compared with senior-year students. They were not exposed to the various complex clinical situations that require critical thinking or the linking of preclinical knowledge and skills to clinical encounters. Students visiting dental clinics have mastered psychomotor skills during their courses. However, these courses did not include other aspects of clinical management, such as communication skills, compassion, and professionalism; therefore, the transition was perceived as a shock to students.

This study demonstrated that there was no significant difference among students enrolled in different years of study in the dental program, considering their perceptions of the transitional stage. This result varies from a recently reported study in which senior-year students had a different perception than students in earlier years of the dental program (de Souza Ferreira et al., 2023). This can be attributed to the clinical program at our college, which operates as a comprehensive dental clinic. Students’ progress to senior clinical years with the same support system and learning environment. The only difference was in the form of experience gained from clinical exposure. Students started their clinical practice in their third year of the program at the first comprehensive clinic. As they move towards senior years, that is, the fourth year of the program, they are exposed to more clinical courses and hence more exposure in the dental clinics and workload.

This study found that the transition of students to dental clinics showed significant influence on learning experience while workload, interaction with patients, and application of skills in clinics did not show significant effects. This is similar to a study conducted at an Australian University, which found that this transition is a complex process and reported similar factors affecting the transition process (Malau-Aduli et al., 2022). In contrast when these variables are controlled, the transition process can be improved. Overall, this study demonstrated that the learning experience of dental students was positively correlated with the application of clinical skills, workload, patient interaction, and transition to clinics. This suggests that these factors play a significant role in shaping perceptions of dental students’ learning experiences. Lazarus & Folkman’s (1984) transactional model of stress and coping emphasizes the dynamic interplay between stressors, individuals’ appraisal of those stressors, and their coping strategies. The transition to clinical training may introduce new stressors such as patient interactions, time pressure, and performance expectations. Effective coping strategies, such as problem-focused coping (e.g., seeking support, time management) and emotion-focused coping (e.g., relaxation techniques, positive reframing), can help students manage stress during this transition. Those who have opportunities to apply their dental skills in a supervised clinical setting, manage their workload effectively, have positive patient interactions, and receive proper orientation during the transition to a clinic are likely to have better overall learning experiences. Transition theory, developed by Meleis et al. (2010) focuses on understanding and facilitating individuals’ transitions through various life stages or roles. The transition from preclinical to clinical training represents a significant role transition for dental students, involving changes in identity, responsibilities, and relationships. Factors such as support from peers, faculty, and mentors as well as clear expectations and guidance can facilitate a smoother transition.

Dental educators should consider these factors when designing and implementing clinical training programs while focusing on creating a supportive and conducive learning environment that enhances the clinical skills, patient interactions, and overall learning experiences of students. The results also imply that other factors may have a stronger influence on the CGPA of students, and that these variables are not reliable measures for predicting academic performance (Table 8). This study has certain limitations, such as the relatively small sample size and potential autocorrelation in the residuals of the model. Further research with larger sample sizes and a more refined methodology is warranted to obtain more robust results and to validate the findings of this study. Furthermore, the participants shared their experiences, which were limited to a single learning environment, faculty, or curriculum. Another limitation of the study was the possible recall bias, especially of senior-year participants whose transition experiences were more than 1 year old. The identification of significant findings, especially through regression analyses, provides valuable insights into the factors that influence students’ experiences during the transition from preclinical to clinical training. For example, regression analyses may reveal the extent to which variables such as self-efficacy, mentorship availability, and stress management strategies predict students’ perceived preparedness for clinical practice. Understanding these relationships can inform targeted interventions and support mechanisms to enhance students’ transitional experiences and overall well-being. We acknowledge the potential biases introduced by survey methodology or sample selection, which may affect the generalizability of our findings. For instance, self-selection bias may occur if only students who are particularly motivated or dissatisfied choose to participate in the survey, thereby skewing results. Additionally, a response bias may arise if participants provide socially desirable responses or if certain groups of students are overrepresented or underrepresented in the sample. Biases introduced by the survey methodology or sample selection may limit the generalizability of our findings to a broader population of dental students or other educational contexts. It is important to acknowledge these limitations and to interpret the results within the context of the study population and methodology.

Theoretical implications: The themes identified in this study can be related to social cognitive theory by Bandura (1989) which states that learning takes place through the interaction of learners with peers and the environment. Learning occurs when personal, environmental, and behavioral factors interact with each other in the learning environment. Another theory which can be related to our findings is cognitive load theory which can affect the learning process in the transition phase (Leppink, 2017). This theory explains that, at the beginning, learners have limited memory (working memory) which is affected by the load placed on the learner at that point in time. In this case, the point of reference is the transition stage, in which the learner is overburdened by the increased cognitive load of the new learning environment.

Recommendations

This study highlights the challenges and experiences of dental students during their transition from preclinical to clinical training. These findings highlight the importance of proper orientation programs, workload management, patient interaction, the application of clinical skills, and overall learning experience in shaping the perception of students in their clinical training. These insights can inform dental education programs and provide interventions to improve the clinical training experiences of dental students. This study has several practical implications for dental educators and institutions. Dental schools should invest in comprehensive orientation programs to provide students with clear expectations and guidance during the transition to clinical training. This can be done before the start of the clinical rotations. The merits of orientation programs for clinics have been established in literature. These programs will help settle learners in a new environment, reduce their anxiety, and motivate them to adapt. The orientation program should be incorporated into the academic plan, and a designated co-coordinator should be appointed to ensure formal implementation. This requires support from the administration of the institutes. Time management training and support systems should also be available to help students effectively manage their workload and reduce their stress levels. A mentorship program can be started in dental schools, where a clinical mentor is allocated to a small group of students. The role of this clinical mentor can provide guidance in clinics as well as allocate patients to them, keeping in mind their level of experience and complexity of the case. This will help gradually increase the cognitive load with easier tasks to be performed earlier and more difficult ones later. This will also help students develop their professional competence. Clinical faculties should also provide adequate supervision and support during clinical training to enhance the application of clinical skills and overall learning experience of students. Additionally, dental curricula should be aligned to ensure that students are adequately prepared with relevant knowledge and skills for clinical practice.

Moreover, communication skills and professional training should be integrated into the dental curriculum to ensure that students are sufficiently prepared to interact with their patients in a competent and compassionate manner.. Dental educators should continuously assess and address the challenges and needs of dental students during the transition to clinical training. By providing supportive and conducive learning environments, dental schools can help students develop necessary skills, knowledge, and professionalism to become competent and confident.

Conclusion

This study aimed to explore dental students’ perceptions regarding the transition from preclinical to clinical training in dentistry, which demonstrated that it is a complex process and that there are multiple factors that affect this process. Increased workload, knowledge gaps, and preparation for clinics can affect this transition. Introducing a clinical orientation, mentoring programs, supervision, and feedback can help students settle into a new learning environment. The findings of this study have implications for the performance of students in dental clinics as they emphasize the strengths of the existing program and the additional support required for student learning.

Supplemental Information

Supplemental Information 1 Normality test results.

Supplemental Information 2 Dataset.

Additional Information and Declarations

Competing Interests

Author Contributions

Human Ethics

Data Availability

The authors declare that they have no competing interests.

Osama Khattak conceived and designed the experiments, authored or reviewed drafts of the article, and approved the final draft.

Kiran Kumar Ganji conceived and designed the experiments, analyzed the data, authored or reviewed drafts of the article, and approved the final draft.

Azhar Iqbal performed the experiments, authored or reviewed drafts of the article, and approved the final draft.

Mosa Altassan analyzed the data, prepared figures and/or tables, and approved the final draft.

Farida Habib Khan performed the experiments, prepared figures and/or tables, and approved the final draft.

Rabia Anis performed the experiments, analyzed the data, prepared figures and/or tables, and approved the final draft.

The following information was supplied relating to ethical approvals (i.e., approving body and any reference numbers):

Local Committee for Bioethics, Jouf University.

The following information was supplied regarding data availability:

The raw data are available in the Supplemental Files.

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
