# Peer review of "Beyond labs: unveiling dynamics of dental students’ transition from pre-clinical to clinical training in a Saudi dental school"

_PeerJ, doi:10.7717/peerj.18019_

## Round 0.1 · original submission · Major Revisions

There were issues with the study design, methodology, and consistency of writing.

**Language Note:** The Academic Editor has identified that the English language must be improved. PeerJ can provide language editing services - please contact us at [email protected] for pricing (be sure to provide your manuscript number and title). Alternatively, you should make your own arrangements to improve the language quality and provide details in your response letter. – PeerJ Staff

Reviewer 1 ·

Basic reporting

The article was well designed, however, the idea was not new as there were previous similar studies.

The findings were descriptive and

What makes this study different than other studies?, is it a replication with a different population?

Experimental design

In the study design, it was stated in para 116 that "this cross-sectional descriptive study was conducted at College of Dentistry, Jouf University that included the collection and analysis of quantitative data through a survey, held at the end of the 3rd academic year of Bachelor of Dental & Oral Surgery (BDS) program, allowing students to reflect on their experiences of previous completed courses."

However, in participants section, para 125, the statement is contradicting "The study population 126 comprised 65 students who were in their third, fourth, and fifth year of the BDS program".

Validity of the findings

1) The statistic of the results was wrongly presented in para 208-2011 "The Mann-Whitney U test demonstrated that the U-value was [31.56-43.24] and the p-value was [p>0.05]. The p-value was less than the significance level of 0.05, indicating that there was no significant difference in the variables (transition to clinics) between third- and fourth-year students (Fig 2)."

2) The P was shown to be larger than 0.05 but the explanations said otherwise.

3) There was no comparison oof mean rank responses for variables among third and fifth year students since there was comparison of mean rank responses for variables among third and fourth year students.

4) A table to present the data of the analyses for the variables should be given in this article to validate the reporting in the results. i.e Mann-Whitney U test was not given in table format.

Additional comments

This study highlights the fact that a proper orientation should be given to dental students before starting a clinic phase. Some comments were made on the curriculum design and review to prepare students for this transition, however, the elaboration of this statement may be necessary to make the recommendation clearer. the findings are not new and the conslusion does not indicate a solution to a well-known problem.

Reviewer 2 ·

Basic reporting

Since dental education courses differed world-wide, more details to the course involved in the study may be described,
The result presentation should be more concise. From line 204 to line 210, the presentation were too verbose without appropriate illustration of the results, which may be modified as “The U-value in Mann-Whitney U test was [33.09-44.33](p>0.05). ”Or “no statistical significance was found in the Mann-Whitney U test.”And the “p” should be written in italic.
The resolution of figure 1 is fuzzy which should be modified.

Experimental design

I noticed that the total amount of participants in the survey was 63 students and the final response rate was 70% resulting 44 participants included. Though there was the calculation, the sample size still seemed a little small. The authors may give some explanation why the response rate was low. And whether it is possible to expand the sample size?
Since the total number of participants was 44, the description in results with only the percentage number will not be enough, for example, “60% strongly felt that orientation should be  provided at the start of clinical practice” may be altered to “26 students (59.09%) strongly felt that…”Also, the percentage figure seemed to be calculated with rounding-off method, the original number may be preferred.

Validity of the findings

Though multiple statistical methods were used, more explanation to the statistical index should be included to facilitate readers understanding why these index are important for the study.

Additional comments

A more concise discussion section may be considered.

Reviewer 3 ·

Basic reporting

Theoretical Implications: The discussion could benefit from a more detailed exploration of the theoretical implications of the findings. This might include considerations of educational theories (e.g., adult learning theory, transition theory) or psychological frameworks (e.g., stress and coping models) that could help explain why certain factors are particularly influential during the transition from preclinical to clinical training. This can be added in the introduction.

Practical Implications: While the manuscript does propose recommendations such as orientation programs and mentoring, further elaboration on how these could be practically implemented within dental curricula would be beneficial. For example, discussing potential challenges and strategies for integrating these recommendations into existing programs could offer valuable guidance to educators and administrators. Some dental schools in other parts of the world have already practised this in their curriculum.

Statistical Analysis and Data Presentation
Response Rate and Participant Characteristics; The research paper indicates a response rate of 70%, which is quite high for surveys, adding credibility to the results. However, understanding the demographics of the participants (such as year of study and gender) is vital in assessing how representative the sample is. While it mentions the gender breakdown (58% female, 42% male), delving into details could enhance the analysis, especially if differences in perspectives based on year of study or gender were investigated.

Implementing a Likert scale for survey responses is appropriate for capturing the subtleties in participants' viewpoints and experiences. The paper effectively showcases how responses are distributed across aspects like transitioning to clinics, workload and patient interactions. To enhance clarity, summarizing these distributions in form or utilizing aids like bar charts could make the findings more understandable to readers. The way the table is reported and/or summarized could also be improved in the result section. In terms of Figures and Tables presentation, the manuscript lays out data in tables showing response frequency distributions and results from regression analyses. While informative, incorporating representations could enhance points for better impact. For instance, visual aids, like graphs showcasing stressors or challenges during transitions, could effectively underscore areas warranting intervention.

Regarding regression analysis, the research paper mentions using regression analysis to investigate how factors such as transitioning to settings and workload relate to learning experiences. The details provided in the tables about standardized coefficients, significance levels and the Durbin Watson statistic indicate an interplay. It would be beneficial to have a discussion focusing on the practical implications of these findings. Specifically delving into the significance of predictors and examining the overall model, e.g., R square values) would offer a clearer understanding of factors influencing student learning experiences.

Experimental design

The information in the study design and study participants is not consistent:
Under study design:
This cross-sectional descriptive study was conducted at the College of Dentistry, Jouf University, that included the collection and analysis of quantitative data through a survey held at the end of the 3rd academic year of the Bachelor of Dental & Oral Surgery (BDS) program,……

The study population comprised 65 students who were in their third, fourth, and fifth year of the BDS program; because two students dropped out of the program in their fourth year, 63 students were actually to complete the clinical courses.

Then later…

The study population included a mix of students, including those who recently completed the pre-clinical course (third-year students) and those who completed the courses nearly two years ago but had more than two years of clinical experience (fifth-year students).

A clearer description of the study population and respondents needs to be spelled out.

More detailed information about the sample selection process, including inclusion and exclusion criteria, would enhance the transparency and replicability of the study.


Data collection:

Methodological Details: Provide more details on the validation process of the questionnaire, including the rationale behind the selection of specific domains and items. Include a thorough description of the survey instrument used, including its development, validation process, and reliability testing.

The authors did mention about validating the adopted questionnaire from Prince et al. However, the questionnaire was originally in Dutch and was translated into English for publication. Then, the English version (I assume) was adopted in the local setting. (Was it translated into Arabic or English?) And if so, how was it translated into the local language and translated back into English? The teaching and learning activities of the dental programme at the institution are being carried out in what language?

This process needs to be described in detail for it to be said to be a validated questionnaire. And what type of validation was being conducted?

Data analysis

The data were imported into Microsoft Excel, and statistical analyses were performed using the Statistical Package for the Social Sciences (SPSS) IBM Corp., which was released in 2017, Version 25.0. Armonk, NY: USA. A descriptive statistical analysis was conducted using SPSS v.25 (IBM Corp., Armonk, NY, USA) to calculate the mean and frequency of the ratings.

The statement on SPSS was repeated in the paragraph. Suggest rewording.
In the result section, a few inferential statistics were reported. So, the types of statistical tests should also be mentioned here under data analysis.

Validity of the findings

Results

Numbers starting with 0 should be written in full
0.001 not .000

Use charts to summarize findings visually for easier understanding.

Statistical Analysis and Data Presentation
Response Rate and Participant Characteristics; The research paper indicates a response rate of 70%, which is quite high for surveys, adding credibility to the results. However, understanding the demographics of the participants (such as year of study and gender) is vital in assessing how representative the sample is. While it mentions the gender breakdown (58% female, 42% male), delving into details could enhance the analysis, especially if differences in perspectives based on year of study or gender were investigated.

'The number of respondents are not consistent: In the methods section:
The study population comprised 65 students who were in their third, fourth, and fifth year of the BDS program; because two students dropped out of the program in their fourth year, 63 students were actually to complete the clinical courses."

However, the number of responses reported in the tables = 71.

Implementing a Likert scale for survey responses is appropriate for capturing the subtleties in participants' viewpoints and experiences. The paper effectively showcases how responses are distributed across aspects like transitioning to clinics, workload and patient interactions. To enhance clarity, summarizing these distributions in form or utilizing aids like bar charts could make the findings more understandable to readers. The way the table is reported and/or summarized could also be improved in the result section. In terms of Figures and Tables presentation, the manuscript lays out data in tables showing response frequency distributions and results from regression analyses. While informative, incorporating representations could enhance points for better impact. For instance, visual aids, like graphs showcasing stressors or challenges during transitions, could effectively underscore areas warranting intervention.

Regarding regression analysis, the research paper mentions using regression analysis to investigate how factors such as transitioning to settings and workload relate to learning experiences. The details provided in the tables about standardized coefficients, significance levels and the Durbin Watson statistic indicate an interplay. It would be beneficial to have a discussion focusing on the practical implications of these findings. Specifically delving into the significance of predictors and examining the overall model, e.g., R square values) would offer a clearer understanding of factors influencing student learning experiences.

Discussion:

Emphasize the novelty of your research by comparing your findings with previous studies and discussing any new insights or contradictions your study presents.

Provide details on how significant findings, especially from regression analyses, impact students' experiences. Address biases introduced by survey methodology or sample selection. How they may impact the generalizability of findings.

Additional comments

Some suggestions for improvements;

Be consistent in the use of the terms:
Preclinical and pre-clinical.
widespread and widespread

Be consistent with UK or US spelling.

The title can be more specific to the region.

Abstract:

Line 34
Background:_Studies have shown that dental students often feel unprepared for clinical practice after undergraduate courses. The transition from preclinical to clinical training can lead to difficulties that require further attention.

Remove _
Background: Studies have shown that dental students often feel unprepared for clinical practice after undergraduate courses. The transition from preclinical to clinical training can lead to difficulties that require further attention.

Line 44
Add ‘.’ After landscape

Likert scale responses were analyzed using SPSS v.25, revealing insights into clinical workload, patient interaction, and learning experiences, offering valuable perspectives on dental education's evolving landscape

Change to

Likert scale responses were analyzed using SPSS v.25, revealing insights into the clinical workload, patient interaction, and learning experiences, offering valuable perspectives on dental education's evolving landscape.

References:
9. Drovandi A, Adu M, Alele F, Jones K, Knott G, Malau-Aduli B. Factors influencing the transition of pharmacy students from pre-clinical to clinical years at university. Pharmacy Education. 2020:336-45.
Missing volume number

17. Moore R, Molsing S, Meyer N, Schepler M. Early Clinical Experience and Mentoring of Young Dental Students-A Qualitative Study. Dent J (Basel). 2021;9(8).
Missing page number

18. Divaris K, Barlow PJ, Chendea SA, Cheong WS, Dounis A, Dragan IF, et al. The academic environment: the students' perspective. Eur J Dent Educ. 2008;12 Suppl 1:120-30.
Suppl 1 – not in bracket

19. Van Merriënboer JJG, Sweller J. Cognitive Load Theory and Complex Learning: Recent Developments and Future Directions. Educational Psychology Review. 2005;17(2):147-
The name of the journal was written in full – not consistent with the others

20. Jones JA, Snyder JJ, Gesko DS, Helgeson MJ. Integrated Medical-Dental Delivery Systems: Models in a Changing Environment and Their Implications for Dental Education. Journal of Dental Education. 2017;81(9):eS21-eS9.
The name of the journal was written in full – not consistent with the others

Annotated reviews are not available for download in order to protect the identity of reviewers who chose to remain anonymous.

---

## Round 0.2 · Minor Revisions

More refinement in your reporting especially statistical analyses and results section are needed.

Reviewer 2 ·

Basic reporting

In line 228 and 231, the data may be presented as “33.09 to 40.33, (p>0.05) indicating…”

Experimental design

no comment

Validity of the findings

no comment

Additional comments

no comment

Reviewer 3 ·

Basic reporting

The language is more polished and professional in terms of the research's basic reporting. The introduction is now more detailed and provides a clearer context, including references to relevant literature. Figures and tables are better integrated and described.

Language and Style:
Continue to refine the language for clarity and conciseness.

Experimental design

The methodology is described in more detail, making it easier to understand and replicate.
The findings are presented in more depth, including a discussion of their implications for dental education.

Justify the choice of statistical methods more thoroughly.

Some additional suggestions:

Mann-Whitney U Test:
This is a non-parametric test used to compare differences between two independent groups when the data does not necessarily follow a normal distribution. It is often used as an alternative to the independent t-test when the assumption of normality is not met.

Multiple Regression Analysis:
This analysis assumes that the residuals (errors) are normally distributed.

Therefore, it is essential to check for the normality of the residuals when performing regression analysis.

Evidence of Normality Testing
To validate the appropriateness of the statistical tests, I would suggest providing evidence of normality testing. This may include:

Descriptive Statistics: Summary statistics for each variable to get an initial sense of their distribution.
Histograms and Q-Q Plots: Visual methods to assess the distribution of the data.
Normality Tests: Such as the Shapiro-Wilk test or Kolmogorov-Smirnov test.

Justify the Use of Non-Parametric Tests:
Given that the Mann-Whitney U test is non-parametric, it is appropriate for the data if it does not follow a normal distribution. However, a brief justification for choosing this test over parametric alternatives should be provided. Providing this additional information will strengthen the methodological rigor of the study and ensure that the statistical analyses are appropriately justified.

Validity of the findings

Reporting of result:

Suggestions for Improvement in the reporting of results.

Intro:
Begin with an introduction to the results:
Begin with a brief overview of the results to orient the reader. Explain what the main findings will cover before diving into the statistical details.

Example:
"This section presents the findings of our study on the transition from pre-clinical to clinical training in dental education. The analysis covers the relationships between the learning experience and various predictors, including clinical skills application, workload, transition smoothness, and patient interaction."

Descriptive Statistics:
Provide a summary of the descriptive statistics first to set the context for the regression analysis.

Example:
"A total of 44 students participated in the survey, with a response rate of 70%. The sample comprised 58% females and 42% males. Key descriptive statistics for the main variables are summarized in Table X."

Regression Analysis:
Report the regression results more systematically. Explain what each statistic means in a straightforward manner.

For example, I’m referring to the reporting in line 202:

The R value was 0.630, indicating a moderately positive relationship between learning experience and application of clinical skills, workload, transition to clinics, and patient interaction. The adjusted R square value was 0.361, indicating that 36.1% of the variance in the learning experience can be explained by predictor variables, such as the application of clinical skills, workload, transition to clinics, and patient interaction. The Durbin-Watson statistic ranges from 0 to 4, with values near 2 indicating no significant autocorrelation. The Durbin-Watson value was 1.850, suggesting positive autocorrelation.


Suggestion:
"The multiple regression analysis showed that the predictors collectively explained 36.1% of the variance in the learning experience (Adjusted R² = 0.361). The overall model was significant (F = X.XX, p < 0.05). The R value of 0.630 indicates a moderately positive relationship. The Durbin-Watson statistic of 1.850 suggests no significant autocorrelation, ensuring the reliability of our regression model."

Individual Predictors:
Discuss the contribution of each predictor variable in more detail.

Example:
"Among the predictors, 'Transition to Clinics' had a significant positive effect on learning experience (β = 0.292, p = 0.012), indicating that smoother transitions were associated with better learning outcomes. However, 'Workload' (β = -0.203, p = 0.393) and 'Patient Interaction' (β = 0.443, p = 0.168) were not significant predictors, suggesting that other factors may be more critical in influencing the overall learning experience."

Tables and Figures:
Refer to tables and figures in the text to support your findings.

Example:
"Table 6 presents the detailed regression coefficients for each predictor variable. As shown, the 'Transition to Clinics' had the highest standardised coefficient, emphasising its importance in the student's learning experience."

Example Revised Results Section

Results

This section presents the findings of our study on the transition from pre-clinical to clinical training in dental education. The analysis covers the relationships between the learning experience and various predictors, including clinical skills application, workload, transition smoothness, and patient interaction.

Descriptive Statistics

A total of 44 students participated in the survey, with a response rate of 70%. The sample comprised 58% females and 42% males. Key descriptive statistics for the main variables are summarised in Table 1.

Regression Analysis

The multiple regression analysis showed that the predictors collectively explained 36.1% of the variance in the learning experience (Adjusted R² = 0.361). The overall model was significant (F = X.XX, p < 0.05). The R-value of 0.630 indicates a moderately positive relationship. The Durbin-Watson statistic of 1.850 suggests no significant autocorrelation, ensuring the reliability of our regression model.

Individual Predictors

Among the predictors, 'Transition to Clinics' had a significant positive effect on learning experience (β = 0.292, p = 0.012), indicating that smoother transitions were associated with better learning outcomes. However, 'Workload' (β = -0.203, p = 0.393) and 'Patient Interaction' (β = 0.443, p = 0.168) were not significant predictors, suggesting that other factors may be more critical in influencing the overall learning experience. The standardised coefficient (beta) for 'Transition to Clinics' indicates that a one-unit increase in this variable is associated with a 0.292 standard deviation increase in the learning experience, holding other predictors constant (Table 7).

Tables and Figures

Table 6 presents the detailed regression coefficients for each predictor variable. As shown, the 'Transition to Clinics' had the highest standardised coefficient, emphasising its importance in the student's learning experience. Table 8 summarises the multiple regression analysis for cumulative GPA, indicating that the predictors are not strong determinants of GPA, with low R² values.


Upon reviewing the discussion section, I noticed a potential discrepancy between the results and the discussion. Specifically, the statement on line 327 reads:

"This study found that the transition of students to dental clinics was significantly influenced by their workload, interaction with patients, and application of skills in clinics."

However, according to the results, only the "Transition to Clinics" variable showed a statistically significant influence (β = 0.292, p = 0.012), while workload, interaction with patients, and application of skills in clinics did not show significant effects.

(From their results)
Transition to Clinics: Had a significant positive effect on learning experience (β = 0.292, p = 0.012).
Workload: Not significant (β = -0.203, p = 0.393).
Patient Interaction: Not significant (β = 0.443, p = 0.168).
Application of Clinical Skills: Not significant (β = -0.002, no p-value mentioned, but implied to be not significant).

To ensure accuracy and consistency, perhaps they can revise this statement in the discussion section to reflect the actual findings reported in the results.

Annotated reviews are not available for download in order to protect the identity of reviewers who chose to remain anonymous.

---

## Round 0.3 · accepted · Accept

Dear Authors,

Thank you for your diligent efforts in addressing the reviewers' comments and making the necessary revisions to your manuscript. We have carefully reviewed the updated version and are pleased to inform you that your article has been accepted for publication.

Congratulations on your successful submission, and we look forward to seeing your work published.

Best regards,